# Efficacy and Safety of Placental Extract on Menopausal Symptoms: A Systematic Review

**DOI:** 10.3390/nu17243857

**Published:** 2025-12-10

**Authors:** Sára Papp, László Tűű, Katalin Nas, Zsófia Telkes, Lotti Keszthelyi, Márton Keszthelyi, Nándor Ács, Szabolcs Várbíró, Marianna Török

**Affiliations:** 1Department of Obstetrics and Gynaecology, Semmelweis University, 1082 Budapest, Hungary; papp.sara@semmelweis.hu (S.P.); telkeszsofi@gmail.com (Z.T.); keszthelyi.lotti.lucia@semmelweis.hu (L.K.); keszthelyi.marton@semmelweis.hu (M.K.); varbiroszabolcs@gmail.com (S.V.); 2School of PhD Studies, Semmelweis University, 1085 Budapest, Hungary; tuul68@gmail.com; 3EndoCare Institute, Endocrinology Center, 1095 Budapest, Hungary; drnaskatalin@yahoo.com; 4Department of Obstetrics and Gynecology, University of Szeged, 6725 Szeged, Hungary

**Keywords:** menopausal symptoms, placenta extract, Kupperman Menopausal Index, Simplified Menopausal Index, Menopausal Rating Scale, hot flashes, wrinkles

## Abstract

**Background:** Menopause affects every woman worldwide, with varying degrees of severity. In addition to traditional treatments such as hormone replacement therapy, there is also a growing interest in alternative treatments. One possible way to address this need is through the use of placenta extracts. This systematic review is the first to evaluate the efficacy of placental extracts in randomized controlled trials (RCTs). **Methods:** A systematic search of three databases (MEDLINE, Scopus, Embase) identified studies on placental extract treatment of menopausal symptoms in women, yielding 272 records, with 11 eligible studies. **Results:** Menopausal severity scores (Kupperman Menopausal Index, Simplified Menopausal Index, Menopausal Rating Scale), somatic and vasomotor symptoms, skin conditions, and certain psychological indicators were significantly improved in the 11 enrolled randomized controlled trials, including perimenopausal and postmenopausal women treated with porcine or human dried, purified placental extract. Placental extract was well tolerated in all studies; no significant side effects or clinically significant laboratory abnormalities were recorded. **Conclusions:** Porcine and human placental extracts appear to offer potential benefits for alleviating menopausal symptoms.

## 1. Introduction

The irreversible termination of ovarian follicular function and the ensuing drop in the production of estrogen and progesterone are hallmarks of the physiological and endocrinological shift known as menopause. It marks the end of the reproductive lifespan and usually happens between the ages of 45 and 55 [1]. Vasomotor symptoms (hot flashes, night sweats), urogenital atrophy, sleep disturbances, mood swings, and metabolic changes are typical clinical signs. Menopause affects every woman worldwide, with varying degrees of severity. The international literature has several studies on the topic of menopause and its symptoms. Menopause and its accompanying symptoms can have a detrimental effect on quality of life and be extremely difficult for both patients and professionals to manage. The symptoms of menopause can widely vary between different ethnic groups [2]. Up to 80–90% of women are said to experience menopausal symptoms, with about one-third of them indicating that the symptoms are severe and negatively impact their quality of life on a daily basis [3]. The severity of menopausal symptoms can be measured using different scores, such as the Menopause Rating Scale (MRS), Greene Climacteric Scale (GCS), Kupperman Index (KMI), and Simplified Menopausal Index (SMI).

The Menopause Rating Scale (MRS) is an 11-item, self-administered questionnaire designed to assess the severity of menopausal symptoms and their impact on quality of life across three aspects: somatic (e.g., hot flashes, joint pain), psychological (e.g., mood, irritability), and urogenital (e.g., sexual problems, bladder symptoms) [4].

The Greene Climacteric Scale (GCS) is a 21-item questionnaire assessing menopausal symptoms across psychological, somatic, vasomotor, and sexual domains. It is mentioned here for reference but was not analyzed in this review [5].

Using a 4-point scoring system (0 = none to 3 = severe), the Kupperman Index (sometimes called the Blatt–Kupperman Menopausal Index) is a questionnaire designed to gauge the intensity of menopausal symptoms; the weighted values are then added up [6].

The Simplified Menopausal Index (SMI) is a widely used menopausal severity score; the SMI is a 10-item self-report questionnaire used to evaluate menopausal symptoms, namely musculoskeletal, vasomotor, and psychoneurological issues. Women use a 4-point severity scale to rank 10 symptoms (e.g., joint pain, mood swings, and hot flashes), with total values ranging from 0 to 100. A gynecological examination may be necessary if the score is 26 or higher, which denotes menopausal symptoms [7,8].

Lifestyle changes, talking or counseling therapies, nutritional supplements, physical activity, and prescription drugs like hormone replacement therapy are possible ways to manage more severe symptoms. Hormone replacement therapy (HRT) is the conventional treatment for menopausal symptoms, offering significant benefits in symptom relief and quality of life, while also presenting potential risks that require an individualized assessment [9]. Dietary supplements are also extensively studied therapeutic options in menopause, and, according to some studies, placenta extract appears to be a promising treatment.

The placenta is a transient organ that aids fetal development by preserving blood supply, eliminating waste, and using placental mesenchymal stem cells (pMSCs) to alter the mother’s immune system. The placenta is rich in nutrients, growth factors, amino acids, hormones, cytokines, and polydeoxyribonucleotides with regenerative, anti-inflammatory, and antioxidant qualities [10]. Placental extract may have therapeutic uses in inflammation, tissue repair [11,12], osteoarthritis [13], and chronic illness according to studies that demonstrate its ability to lower oxidative stress [14], encourage tissue regeneration [15], alleviate fatigue [16], enhance cell viability [17], and lessen menopausal symptoms. An investigation of the efficacy of placental extracts for menopausal symptoms was performed in rat models [18] and has been tested later in humans as well, but, until now, there has never been a comprehensive systematic review comparing multiple studies on the effects of placental extracts on menopausal symptoms.

## 2. Materials and Methods

This study was registered in PROSPERO under the reference number CRD420251166797. This review follows the PRISMA 2020 [19] guidelines and the Cochrane Handbook for Systematic Reviews of Interventions (version 6.3).

### 2.1. Qualification Standards

Research involving patients receiving placental extract treatment for menopausal symptoms was included in our study.

### 2.2. Data Collection

The studies were selected from the MEDLINE (via PubMed), Embase, and Scopus databases. The search key was as follows: (menopause OR perimenopausal OR postmenopausal OR climacteric) AND (placental extract OR placenta therapy OR placenta-derived treatment). There were no limitations on language or any other aspects. Although no language restrictions were applied during the search, only English-language articles were identified and included in this review. Consequently, translations and risk of bias assessments for non-English articles were not required. Articles published between 2004 and 2025 were considered in the search intentionally to ensure a comprehensive 20-year overview of the available literature on placental extract therapies. A total of 21 articles were discovered in Scopus, 267 in Embase, and 63 in PubMed based on the given criteria.

### 2.3. PICO Criteria

Our review question and PICO criteria for our research were the following:

Is placental extract a safe and effective treatment for menopausal symptoms?

P—Menopausal women;I—Placental extract treatment;C—Placebo or sham treatment;O—Change in menopausal symptom severity and safety profile.

### 2.4. Data Selection

Selection was performed by using PICO criteria following a methodical database search and the subsequent elimination of duplicates. The EndNote X9 reference management program (Clarivate Analytics, Philadelphia, PA, USA) was used to manage references and remove duplicates. Two independent authors (S.P. and Z.T.) individually screened the publications based on title, abstract, and full text.

### 2.5. Selection of Title and Abstract

We included studies involving menopausal women who received placental extract therapy for any menopausal symptoms, whether they were randomized or not. We excluded reviews, letters, commentaries, and protocols and other publications lacking original research data. We initially screened studies regardless of their design, but only randomized controlled trials met the criteria for final synthesis.

### 2.6. Selection of Full Text

We included studies that used the same outcome measurement units. Studies that did not fit the PICO framework or reported outcomes in incompatible formats were excluded.

### 2.7. Procedure for Gathering Data

Data were independently entered into an Excel spreadsheet (Office 365, Microsoft 16.86, Redmond, WA, USA) by two authors (S.P. and Z.T.). We gathered the following information from the eligible papers: demographics, treatment details, study type, study design, first author, year of publication, and outcome data (primary outcome: SMI score, KMI score, somatic symptoms, vasomotor symptoms; secondary outcome: aesthetics, psychological symptoms).

### 2.8. Assessment of Bias

Two reviewers (S.P. and Z.T.) independently assessed the risk of bias in the results using Risk of Bias 2, an updated risk of bias instrument for randomized trials.

The majority of trials, which were classified as “low risk” or “some concerns,” showed a low or moderate risk of bias overall. Every category was evaluated independently, including the randomization procedure, deviations from intended interventions, missing outcome data, outcome assessment, and selection of reported results. The summed data show generally good methodological quality across trials (e.g., 63.6% for adherence to interventions and 90.9% low risk for randomization). Risk of bias evaluations are available in Appendix A.

## 3. Results

### 3.1. Included Studies

A total of 351 articles from the three databases matched the search key. After removing duplicates, 233 articles were screened. According to the titles and abstracts, we excluded 210 studies and included 23. Furthermore, 12 articles were excluded during the full-text assessment. Overall, 11 studies were included in the systematic review. The summary of the selection process is shown in Figure 1.

### 3.2. Basic Characteristics of Included Studies

Table 1 lists the features of studies selected for the systematic review, along with the treatment characteristics of the studies included. From the 11 articles, 988 patients were involved. All 11 studies were randomized controlled trials. The eligibility criteria included studies screening peri-/postmenopausal or climacteric women with any symptoms treated with porcine or human placental extract.

### 3.3. Primary Outcomes

#### 3.3.1. Menopausal Severity Scores

Out of the 11 studies, 8 analyzed menopausal severity scores. In three articles, the Kupperman Menopausal Index score was examined [20,21,22], in another three, SMI scores were examined [15,23,24], and in two articles the MRS [25,26] was examined.

##### Kupperman Menopausal Index

The Kupperman Menopausal Index is a questionnaire on menopausal symptoms consisting of a total of 11 items and is used to measure the severity of climacteric (menopause-related) complaints. The KMI score was examined in three of the involved studies, including 349 women altogether [20,21,22]. After just 4 weeks of human placenta extract treatment, the KMI decreased by an average of 12 points (the placebo control group decreased by an average of 7 points, *p* = 0.012) [20]. Furthermore, with the oral administration of 400 mg of porcine placental extract daily for 12 weeks, the reduction in the KMI score was significantly greater in early menopausal women (less than 3 years since menopause) and in women with a BMI ≥ 23 kg/m^2^, showing an average decrease of 17–18 points compared with an 11-point decrease in the placebo group (*p* < 0.05) [21]. The third study compared two human placenta extracts, Melsmont and Unicenta. After 12 days of subcutaneous injections (6 injections over 2 weeks), both placenta extracts reduced the KMI to a similar extent (the BMI decreased by an average of 14 points) [22]. Table 2 summarizes the studies dealing with KMI scores.

##### Simplified Menopausal Index

Three studies examined changes in SMI scores in response to placenta extract [23,24]. The daily consumption of 300 mg of porcine placental extract for 12 weeks reduced the SMI score by an average of 14 points (in the placebo control group, the SMI decreased by an average of 6 points) [23]. A 24-week treatment with 1050 mg and then 2100 mg of porcine placenta extract per day reduced the SMI by an average of 30 points. Furthermore, 4 weeks after the end of treatment, the SMI score remained low [24]. In the third study, compared to the other two, PPE was administered for a shorter duration—only 4 weeks versus 12 and 24 weeks. In this study, no significant difference was found in the SMI score compared to the placebo group. However, the skin barrier function (transepidermal water loss) significantly decreased after 4 weeks of porcine placenta extract treatment in this study [15]. Table 3 summarizes the studies dealing with SMI scores.

##### Menopause Rating Scale

To analyze the severity of menopause, a Korean study used the MRS (Menopause Rating Scale) scale and examined the effects of 8 weeks of human placenta extract [25]. The placental treatment group showed a significantly greater reduction in MRS at 8 weeks compared to the placebo group (*p* = 0.033). In another study, 9 weeks of human placenta extract treatment did not significantly reduce MRS values [26].

#### 3.3.2. Somatic Symptoms

Five articles [21,22,23,25,26] examined the somatic symptoms that may occur in menopausal women from among the selected studies. In four of these five studies, porcine placental extract was administered to the study group for different periods of time. In the two articles of Koike et al. [27,28] and in the trial of Kong et al. [25], the Visual Analog Scale (VAS) for shoulder stiffness and knee pain was investigated. In the article by Kitanohara, somatic symptoms, such as “easy fatigability”, “shoulder stiffness”, lumbago, or joint pain, were analyzed as part of the SMI score. A total of 324 patients were examined across the five articles. In all five studies, the somatic symptoms of the observed women were significantly reduced. Shoulder stiffness and knee pain decreased, even in hormone therapy-resistant cases [27,28]. Table 4 summarizes the studies dealing with somatic symptoms.

#### 3.3.3. Vasomotor Symptoms

Five of the involved studies considered vasomotor symptoms. One of the studies focused exclusively on hot flashes [26]. After 9 weeks of human placental extract treatment, the hot flash score (HFS) decreased significantly, but there was no difference between the treatment and placebo groups. One month after the treatment, the human placental extract treatment group showed a greater reduction in HFS compared to the control group; the score increased in the control group [26]. Four studies’ vasomotor symptoms were analyzed as a secondary endpoint in addition to menopausal severity scores [20,22,23,24]. Vasomotor symptoms significantly improved in the placental treatment group, with scores decreasing from 10.30 to 6.13 after 4 weeks, compared to a smaller reduction in the placebo group from 9.45 to 8.34 (*p* < 0.001) [20]. In another study, both human placental extract treatment groups (Melsmon and Unicenta) demonstrated improvements in the frequency of hot flashes during the day, the frequency of hot flashes at night, the total score of hot flashes during the day, and the score of hot flashes during the night that were statistically significant [22]. Furthermore, the treatment with porcine placental extract significantly reduced hot flashes by 70.2% at 24 weeks. The effect sustained a 63.4% reduction four weeks after treatment cessation compared to baseline [24]. Table 5 summarizes the studies dealing with vasomotor symptoms.

### 3.4. Secondary Outcomes

#### 3.4.1. Aesthetics

An open-label, randomized, controlled study enrolled 44 climacteric women with mild or few menopausal symptoms to evaluate the effect of porcine placental extract on wrinkle width by using SDSNA and retrospectively compared the wrinkle widths among the groups. The oral administration of porcine placental extract represented a potential therapeutic option for improving fine wrinkles under the eye in climacteric women [29]. A randomized, double-blind, placebo-controlled study evaluated the effects of oral porcine placenta extract on skin hydration, transepidermal water loss, and elasticity in climacteric women, finding significant improvements in arm skin transepidermal water loss and elasticity parameters in the treatment group compared to a placebo [15]. Table 6 summarizes the studies dealing with aesthetics.

#### 3.4.2. Psychological Symptoms

The SMI subscore involved psychological symptoms such as “insomnia”, “easy excitability or irritability”, “worry about self-depression”, and “headache, vertigo or nausea”. Psychological symptoms significantly improved within the porcine placental group at both 8 and 12 weeks compared to baseline, though no statistically significant differences were found between the PPE and placebo groups [23]. Kong et al. found that psychological symptoms improved in both the human placenta extract group and placebo groups over the study period, with a greater, though not statistically significant, reduction in the human placental extract group compared to the placebo (*p* = 0.089) [25].

The third study investigated specifically the Japanese version of Zung’s Self-Rating Depression Scale (ZSDS) and the Spielberger State–Trait Anxiety Inventory questionnaires (STAI). The porcine placental extract group showed significant improvements in psychological symptoms, as evidenced by reductions in the ZSDS, STAI-1, and STAI-2 scores and the SMI depression subscale at both 12 and 24 weeks compared with the control group (*p* < 0.01), with these benefits persisting for 4 weeks following treatment discontinuation [24]. Table 7 summarizes the studies dealing with psychological symptoms.

### 3.5. Hormonal Changes and Cardiovascular Effects

Four of our involved articles highlight the hormonal and cardiovascular effects of the usage of placental extract [22,23,25,26]. In the articles of Kim et al. and Choi et al., human placental extract caused no hormonal stimulation; estradiol and FSH levels showed no significant changes, indicating that the therapy does not act as systemic hormone replacement. Cardiovascular safety parameters, including blood pressure, pulse, and laboratory markers, remained stable throughout treatment. Similarly, the porcine placental extract in the study of Kitanhora et al. showed no change in serum estradiol or FSH and no cardiovascular events. However, in the study of Kong et al., the 17 beta-estradiol level was significantly increased at 8 weeks in the placental treatment group compared to the placebo group [25].

### 3.6. Adverse Events, Side Effects

Adverse effects were monitored through patient-reported unpleasant events, clinical laboratory tests (lipid profiles, renal and liver function, estradiol, and FSH levels), and physical examinations with vital signs at follow-up visits. These findings were then compared with control or placebo groups to determine whether adverse effects were more common or severe. All cited studies consistently report no significant adverse effects or clinically meaningful laboratory abnormalities with placental extract use in the studied time frames (8–24 weeks).

## 4. Discussion

This study is the first systematic review to examine the efficacy of placental extract on menopausal symptoms. Our goal was to present an assessment of the efficacy of the placental extract and its side effects among menopausal women. Up to 80–90% of women are said to experience some menopausal symptoms during middle age, with about one-third of those who do indicating that the symptoms are severe and negatively impact their daily quality of life [30]. To alleviate these symptoms, the current approaches are hormonal replacement therapies. In addition to well-known alternative therapies, such as exercise and diet, our systematic review suggests that placenta extracts may also be effective in alleviating menopausal symptoms.

Menopausal symptoms are commonly assessed by menopause severity scores such as KMI, SMI, and MRS [31,32,33,34,35].

The Kupperman Index is an eleven-item scale which examines the severity of menopause; vasomotor, paraesthesia, sleeplessness, anxiety, depression, vertigo, weakness, arthralgia and myalgia, headaches, palpitations, and formication are among its eleven components [36]. The Kupperman Index can be lowered with physical activity; the findings suggest that middle-aged women who engage in moderate-to-intense physical activity may experience a reduction in menopausal symptoms [37,38]. In our present study, three studies examined the KMI in response to placenta extracts [22]. Kim et al. [22] compared two injectable placental extract products: Melsmon and Unicenta. Melsmon has been used in Japan for about 70 years to alleviate menopausal symptoms. In a placebo-controlled study, Melsmon significantly reduced mental and physical symptoms after two weeks of treatment (placebo, 25% reduction; Melsmon, 77.4% reduction) [39]. Unicenta, also a human placenta extract, has been available in South Korea for about 20 years to alleviate menopausal symptoms. Based on data from Kim et al., Unicenta reduced the KMI to a similar extent compared to Melsmon. In the Unicenta group, the Kupperman Index decreased by 17.26 points (from 30.64 to 13.38), while in the Melsmon group it decreased by 16.92 points (from 31.31 to 14.39) by the 8th visit [22]. Compared to hormone therapy, the placenta extract reduced the KMI to a similar extent as hormone therapy; in conventional oral hormone replacement therapy, the score decreased by 9.45 points (from 24.11 to 14.66) after 4 weeks [31]. Lee et al. (2020) examined the efficacy of the oral administration of porcine placental extract by evaluating the decrease in the KMI [21]. After the admission of PPE for 12 weeks, the KMI dropped more significantly in the porcine placental group than in the placebo group for overweight or obese women with a body mass index (BMI) of 23 kg/m^2^ or higher. After 12 weeks, the KMI dropped more significantly in the porcine placental group among 49 early menopausal women whose menopause lasted less than three years. The baseline KMI of 35.02 decreased by 18.52 points in the porcine placental extract group compared to 11.40 points in the placebo group among overweight or obese women. Among early menopausal women, the reduction was 17.29 points in the PPE group versus 11.29 points in the placebo group after 12 weeks [21]. In contrast, conventional therapy showed a smaller decrease of 13.29 points after 12 weeks. Compared to traditional hormonal therapy, porcine placental extract reduced the KMI to a similar extent as hormone treatment in menopausal women [31]. The last study (Lee et al. 2009), which used the KMI score in our systematic review, administered human placental extract or a placebo for 4 weeks. The human placental extract reported a 12.30-point decrease versus 7.15-points in the placebo group, indicating a significantly greater improvement in the treatment group [20].

SMI is a widely used score in Japan for examining the severity of menopause in middle-aged women. Ten questions made up the SMI, which evaluated somatic (two items: joint pain, shoulder stiffness), psychological (four items: mood, sleep problems, etc.), and vasomotor (four items: hot flashes, chills, etc.) symptoms [40]. The literature mentions cases where the SMI has decreased in women who practiced stretching for 3 weeks [41]. Three studies looked at the SMI’s reaction to placenta extracts in our current investigation. The SMI can also be alleviated through the oral administration of porcine placental extract [23,24]. In one of these studies (Koike et al., 2012) [28], even after suspending administration of the porcine placental extract, the positive effects lasted. However, Nagae et al. show that, if the drug is used for only 4 weeks, the reduction in the SMI score is not significant [15]. Based on the results of our systematic review, it can be seen that both administration types (subcutaneous injection and oral administration) and both placental types (porcine and human) proved to be effective in terms of decreasing both the KMI and SMI scores.

Women in their climacterium often suffer from somatic symptoms such as muscle and joint pain, insomnia, palpitation headache, and vasomotor symptoms like hot flashes. Postmenopausal women treated with hormone replacement therapy or soy-based phytoestrogens are found both to have significantly reduced somatic–vegetative symptoms, such as hot flashes, palpitations, and sleep disturbances, and joint pain compared to controls [42]. Women receiving dietary counseling and resistance exercise experienced significant improvements in vasomotor symptoms (hot flashes, sweating) and somatic complaints (joint and muscle pain, palpitations, fatigue) during the perimenopausal transition [43]. In our study, we wanted to investigate if these symptoms can be lowered by the administration of a placental extract as well. Two of our selected studies focused specifically on knee pain and shoulder stiffness [27,28]. These studies show that porcine placental extract alone significantly reduces shoulder stiffness. In addition to hormone replacement therapy, it was significantly effective in treating knee pain. Other studies took somatic symptoms as one of the subscores for the menopausal severity scores [23,24,25]. As part of the SMI/MRS scores, this subscore decreased as well. Vasomotor symptoms were measured in one of the examined articles using the hot flash score. In patients who were treated with human placenta, the hot flash score significantly decreased [26]. Vasomotor symptoms are part of both SMI and KMI scores. However, in one of the articles, it is highlighted that porcine placental extract significantly improved menopausal symptoms, particularly hot flashes, insomnia, irritability, depression, fatigue, and joint pain [24].

Placental extract preparations (both human and porcine, injectable or oral) appear to be safe and well tolerated in the short term, with no consistent reports of significant adverse effects compared to a placebo. However, long-term safety and rare events remain insufficiently studied, so cautious use and further research are required. Placental extract has not demonstrated negative cardiovascular consequences, in contrast to systemic menopausal hormone therapy started at an older age (>60 years) or more than 10 years following menopause, which is linked to an unfavorable shift in cardiovascular risk [44]; however, the long-term administration of placental extract has not been evaluated yet.

Strengths and Limitations

All of the enrolled studies are randomized, controlled studies, and the systematic review was written about an almost 1000-person population. However, the international literature is not that rich, and all of the studies enrolled are from Asian countries, so the therapy was not tested within different populations. The measurements did not allow us to make a meta-analysis, as there were not enough studies to create the statistics. None of the studies performed a long-term follow-up, so the long-term effects cannot be studied. The Kupperman Index has been widely used in clinical studies of climacteric symptoms, though it has been critiqued due to weighting without statistical justification, overlapping categories, and a lack of coverage for some symptoms such as vaginal dryness or libido loss [45].

Implications for Practice

It is shown that the currently available drugs which contain human or porcine placental extract may serve as an alternative or additional therapy for menopausal symptoms. Placental extract therapy lowered the KMI and SMI scores in six different randomized, controlled studies. Placental extract therapy also seemed to be significantly effective at alleviating specific symptoms such as vasomotor, somatic, or psychological symptoms. Patients with hormone therapy-resistant somatic symptoms also experienced improvements with the additional placental extract therapy, so these kinds of drugs may serve as a possible solution in hormone therapy-unresponsive cases. However, in the articles studied by us, no adverse events nor serious side effects were reported; the usage of placental extract needs to be studied in long-term studies as well.

Implications for Research

It is known that the symptoms of the menopausal transition can differ between populations. As the studies enrolled by us were mostly studies from Asia, it would be advantageous to study the efficacy of placenta extract for the symptoms of different populations, since urogenital symptoms (which is are some of the most common in our country), for example, were not mentioned in the eleven enrolled studies. As the longest study duration is 24 weeks, a long-term follow-up would also be useful to monitor the long-term effects and possible side effects of these drugs.

## 5. Conclusions

With increasing life expectancy, women now spend nearly one-third of their lives in the climacteric period. The symptoms of menopause affect all women, though with varying degrees of severity. It is a great challenge for professionals to find the most effective therapy with the fewest possible side effects for their patients. The menopausal transition may occur with severe symptoms which vary between different populations. The severity of these symptoms is measured using different menopausal severity scores. Therapy for menopausal symptoms has always been a highly researched topic in the international literature. The conventional therapy nowadays is hormone replacement therapy, but several lifestyle therapies are available, as well as physical activity and dietary supplements. Placental extract is used for different regenerative and anti-aging treatments in Asian countries. Some trials showed that both porcine and human placental extract subcutaneously or orally given can be useful in treating menopausal symptoms. Our systematic review is the first to analyze human patients with menopausal symptoms who received a placental extract as an alternative therapy. The extract significantly alleviated the symptoms without any serious adverse events. Placental extract might be a good alternative or additional therapy for mild menopausal symptoms.

## Figures and Tables

**Figure 1 nutrients-17-03857-f001:**
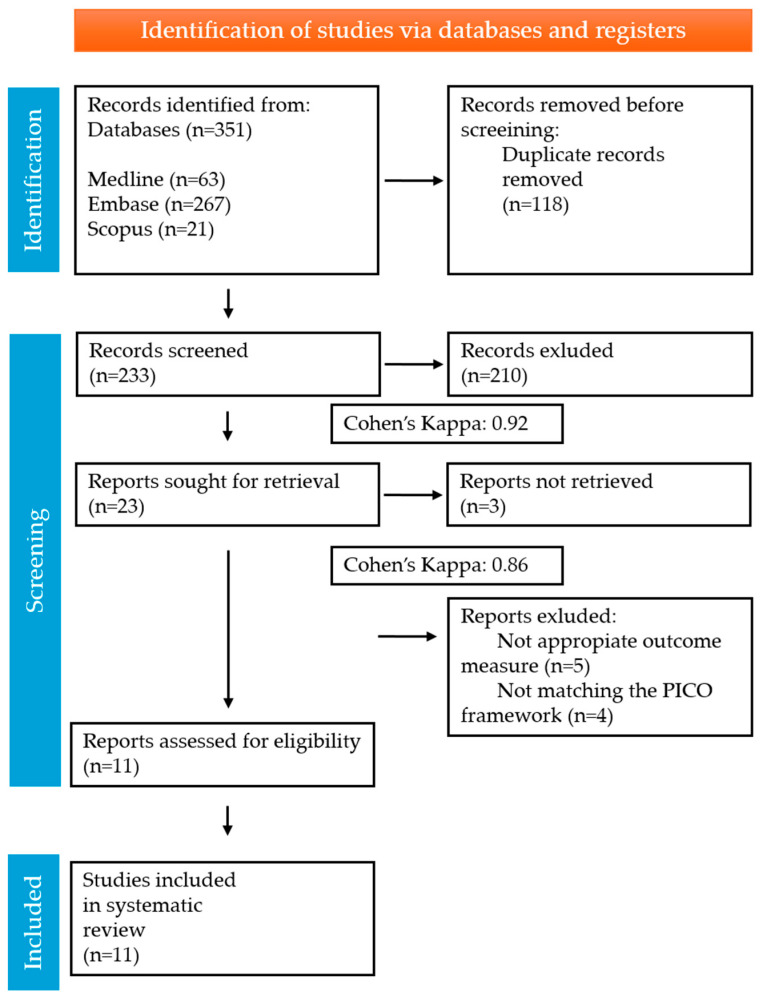
PRISMA Flow Diagram of the screening and selection process.

**Table 1 nutrients-17-03857-t001:** Characteristics of studies included.

Author, Year	Study Design	Number of Patients	Dose of Porcine/Human Placental Extract	Duration of Study	Main Outcomes
Lee et al. (2009) [20]	Randomized, double-blind, placebo-controlled trial	108	100 mg human placental extract 3 times/week s.c.	4 weeks	KMI score, vasomotor symptoms
Lee et al. (2020) [21]	Multicenter, Randomized, double-blind, placebo-controlled trial	100	400 mg porcine placental extract p.o./day	12 weeks	KMI score
Kim et al. (2023) [22]	Randomized, multicenter, double-blind, parallel, non-inferiority clinical study	141	2 mL human placenta extract subcutaneous injection 3 times/week	12 days	KMI score, hormonal changes, vasomotor symptoms
Kitanohara et al. (2017) [23]	Multicenter, randomized, double-blind, placebo-controlled, parallel-group study	50	300 mg porcine placental extract/day p.o.	12 weeks	SMI score, serum estradiol, follicle stimulating hormone, vasomotor, psychological and somatic symptoms
Koike et al. (2013) [24]	Open-label, randomized, controlled study	76	First 12 weeks: 1050 mg porcine placental extract/day p.o.; second 12 weeks: 2100 mg porcine placental extract/day p.o.	24 weeks	SMI score, Zung’s Self-Rating Depression Scale, the Spielberger State–Trait Anxiety Inventory, vasomotor symptoms
Nagae et al. (2020) [15]	Randomized, double-blind placebo-controlled study	20	200 mg porcine placenta extract/day p.o.	4 weeks	SMI score, skin quality parameters
Kong et al. (2008) [25]	Randomized, controlled trial	84	First 2 weeks: 4 mL human placental extract 2 times/week s.c.; second 2 weeks: 2 mL human placental extract 2 times/week s.c.; last 4 weeks: 2 mL human placental extract once a week (total 32 mL)	8 weeks	Menopausal Rating Scale (MRS), Fatigue Severity Scale, Visual Analog Scale to assess the degree of fatigue, 17 beta-estradiol level
Choi et al. (2022) [26]	Randomized, single-blind, placebo-controlled trial	128	2 mL human placental extract subcutaneous injection twice weekly	9 weeks	Hot flash score, the Menopausal Rating Scale, follicle-stimulating hormone (FSH) levels, and estradiol (E2) levels
Koike et al. (2013) [27]	Open-label, randomized, controlled study	66	First 12 weeks: 1050 mg porcine placental extract/day p.o.; second 12 weeks: 2100 mg porcine placental extract/day p.o.	24 weeks	Shoulder stiffness by the Visual Analog Scale
Koike et al. (2012) [28]	Open-label, randomized, controlled study	48	3150 mg porcine placental extract daily p.o. with hormone replacement therapy or estrogen-alone therapy	12 weeks	Knee pain by the Visual Analog Scale
Yoshikawa et al. (2014) [29]	Open-label, randomized, controlled study and retrospective comparison	167	Group 1: 1050 mg porcine placental extract/day p.o.; Group 2: 2100 mg porcine placental extract/day p.o.	12 weeks	Wrinkle widths

**Table 2 nutrients-17-03857-t002:** Kupperman Menopausal Index.

Author (Year)	Study Type	Number of Patients	Porcine/Human Placental Extract	Duration of Treatment	KMI Score Before and After Treatment	Outcome
Lee et al. (2009) [20]	Prospective, randomized, double-blind, placebo-controlled trial	108	100 mg human placental extract subcutaneous injection 3 times/week	4 weeks	36/24	The degree of decrease in the KMI score was significantly greater in the human placental extract group than in the placebo group (−12.30 ± 10.44 vs. −7.15 ± 9.11, *p* = 0.012) after 4 weeks of treatment
Lee et al. (2020) [21]	Randomized double-blind placebo-controlled trial	100	400 mg porcine placental extract per os	12 weeks	35/17–18	KMI significantly decreased after 12 weeks in patients whose BMI was 23 kg/m^2^ or above and in early menopausal women compared to the placebo group
Kim et al. (2023) [22]	Randomized, multicenter, double-blind, parallel, non-inferiority clinical trial	141	2 mL human placenta extract subcutaneous injection 3 times/week	12 days	31/16	The effects of the two human placenta extracts do not differ significantly in terms of KMI

**Table 3 nutrients-17-03857-t003:** Simplified Menopausal Index.

Author (Year)	Study Type	Number of Patients	Porcine/Human Placental Extract	Duration of Treatment	SMI Score Before and After Treatment	Outcome
Kitanohara et al. (2017) [23]	Multicenter, randomized, double-blind, placebo-controlled, parallel-group study	150	300 mg porcine placental extract/day p.o.	12 weeks	36/22	After 12 weeks, the porcine placental extract group’s overall SMI score had improved significantly more than the placebo group’s (*p* = 0.031). The porcine placental extract group’s vasomotor, psychological, and somatic symptoms considerably improved after 12 weeks (*p* < 0.05).
Koike et al. (2013) [24]	Open-label, randomized, controlled study	76	First 12 weeks: 1050 mg porcine placental extract/day p.o.; second 12 weeks: 2100 mg porcine placental extract/day p.o.	24 weeks	50/21	SMI significantly decreased after 24 weeks compared to the control group.
Nagae et al. (2020) [15]	Randomized, double-blind, placebo-controlled trial	20	200 mg porcine placenta extract/day p.o.	4 weeks	50/40	Skin barrier function (transepidermal water loss) significantly decreased after 4 weeks of porcine placental extract treatment. No significant difference was found in the SMI score compared to the placebo group.

**Table 4 nutrients-17-03857-t004:** Somatic symptoms.

Author (Year)	Study Type	Number of Patients	Porcine/Human Placental Extract	Duration of Treatment	Outcome
Koike et al. (2013) [27]	Open-label, randomized, controlled study	66	First 12 weeks: 1050 mg porcine placental extract/day p.o.; second 12 weeks: 2100 mg porcine placental extract/day p.o.	24 weeks	Six capsules of porcine placental extract per day was significantly effective in reducing the Visual Analog Scale
Koike et al. (2012) [28]	Open-label, randomized, controlled study	48	3150 mg porcine placental extract daiy p.o. with hormone replacement therapy or estrogen-alone therapy	12 weeks	Treatment with porcine placental extract was significantly effective in reducing the VAS score for knee pain
Kitanohara et al. (2017) [23]	Multicenter, randomized, double-blind, placebo-controlled, parallel-group study	50	300 mg porcine placental extract/day p.o.	12 weeks	The porcine placental extract group’s vasomotor, psychological, and somatic symptoms considerably improved at 12 weeks (*p* < 0.05)
Koike et al. (2013) [24]	Open-label, randomized, controlled study	76	First 12 weeks: 1050 mg porcine placental extract/day p.o.; second 12 weeks: 2100 mg porcine placental extract/day p.o.	24 weeks	Reductions in the subscale scores of SMI for joint pain: at 24 weeks and at 28 weeks (4 weeks after therapy)
Kong et al. (2008) [25]	Randomized, controlled trial	84	First 2 weeks: 4 mL human placental extract 2 times/week s.c.; second 2 weeks: 2 mL human placental extract 2 times/week s.c.; last 4 weeks: 2 mL human placental extract once a week (total 32 mL)	8 weeks	Fatigue Severity Scale (FSS) score at the end of the study period was significantly decreased from baseline; the mean total Menopause Rating Scale score of the human placental extract group at 8 weeks was significantly lower than that of the placebo group; the mean total Menopause Rating Scale and three subscale scores in the two groups at the end of the study period were significantly lower than the scores at baseline; Visual Analog Scale score in the human placental extract group was significantly decreased at the end 8 weeks

**Table 5 nutrients-17-03857-t005:** Vasomotor symptoms.

Author (Year)	Study Type	Number of Patients	Porcine/Human Placental Extract	Duration of Treatment	Outcome
Choi et al. (2022) [26]	Randomized, single-blind, placebo-controlled trial	128	2 mL human placental extract subcutaneous injection twice weekly	9 weeks	Hot flash score decreased significantly at 9 weeks. One month after 9 weeks, the score of the placental extract group was reduced, but the score increased in the control group.
Kim et al. (2023) [22]	Randomized, multicenter, double-blind, parallel, non-inferiority, clinical study	141	2 mL human placenta extract subcutaneous injection 3 times/week	12 days	Both placental extract groups (Melsmon and Unicenta) showed statistically significant improvements in the frequency of daytime hot flashes, frequency of nighttime hot flashes, total score of daytime hot flashes, and score of nighttime hot flashes. There were no statistically significant differences in the evaluation of facial flashes.
Kitanohara et al. (2017) [23]	Multicenter, randomized, double-blind, placebo-controlled, parallel-group study	50	300 mg porcine placental extract/day p.o.	12 weeks	The porcine placental extract group’s vasomotor, psychological, and somatic symptoms considerably improved at 12 weeks (*p* < 0.05).
Koike et al. (2013) [24]	Open-label, randomized, controlled study	76	First 12 weeks: 1050 mg porcine placental extract/day p.o.; second 12 weeks: 2100 mg porcine placental extract/day p.o.	24 weeks	Reductions in the subscale scores of SMI for joint pain: 63.3% at 24 weeks, 58.3% at 28 weeks (4 weeks after therapy). Significantly effective in reducing hot flashes, insomnia, irritability, depression, fatigue, and joint pain at 12 weeks (*p* < 0.01) and 24 weeks (*p*< 0.01).
Lee et al. (2009) [20]	Randomized, double-blind, placebo-controlled trial	108	100 mg human placental extract 3 times/week s.c.	4 weeks	Vasomotor troubles significantly decreased at 4 weeks.

**Table 6 nutrients-17-03857-t006:** Aesthetics.

Author (Year)	Study Type	Number of Patients	Porcine/Human Placental Extract	Duration of Treatment	Outcome
Yoshikawa et al. [29]	Open-label, randomized, controlled study and retrospective comparison	167	Group 1.: 1050 mg porcine placental extract/day p.o.; Group 2.: 2100 mg porcine placental extract/day p.o.	12 weeks	12 weeks of porcine placental extract treatment caused a significant reduction in wrinkle widths below the eye in subjects treated with three capsules of porcine placental extact per day (*p* < 0.05), as well as in subjects treated with six capsules of porcine placenal extract per day (*p* < 0.01) compared with untreated subjects. Treatment with three capsules of porcine placental extract per day was significantly effective in reducing the wrinkle widths below the eye both at 12 weeks (*p* < 0.05) and 24 weeks (*p* < 0.01) compared with the baseline.
Nagae et al. (2020) [15]	Randomized, double-blind, placebo-controlled study	20	200 mg porcine placenta extract/day p.o.	4 weeks	Simplified Menopausal Index over 4 weeks showed a trend toward higher values in the test group than in the placebo group, which did not reach significance (*p* = 0.079). Arm skin transepidermal water loss was observed, with a significant difference for 4 weeks. Arm skin elasticity was significantly greater in the test group than in the placebo group (*p* = 0.017).

**Table 7 nutrients-17-03857-t007:** Psychological symptoms.

Author (Year)	Study Type	Number of Patients	Porcine/Human Placental Extract	Duration of Treatment	Outcome
Kong et al. (2008) [25]	Randomized, controlled trial	84	First 2 weeks: 4 mL human placental extract 2 times/week s.c.; second 2 weeks: 2 mL human placental extract 2 times/week s.c.; last 4 weeks: 2 mL human placental extract once a week (total 32 mL)	8 weeks	Fatigue Severity Scale score (3.2 ± 1.4) at the end of the study period was significantly decreased from baseline
Kitanohara et al. (2017) [23]	Multicenter, randomized, double-blind, placebo-controlled, parallel-group study	50	300 mg porcine placental extract/day p.o.	12 weeks	The porcine placental extract group’s vasomotor, psychological, and somatic symptoms considerably improved at 12 weeks (*p* < 0.05)
Koike et al. (2013) [24]	Open-label, randomized, controlled study	76	First 12 weeks: 1050 mg porcine placental extract/day p.o.; second 12 weeks: 2100 mg porcine placental extract/day p.o.	24 weeks	Significantly effective in reducing the total SMI, ZSDS, STAI-1, and STAI-2 scores at week 12 and week 24; significantly effective in reducing hot flashes, insomnia, irritability, depression, fatigue, and joint pain at 12 weeks and 24 weeks; reductions in the subscale scores of SMI for irritability, depression: 75.3%, 61.2% at 24 weeks, 60.3%, 51.2% at 28 weeks (4 weeks after therapy); insomnia subscale score reduction: 74.7% at 24 weeks, 57.9% at 28 weeks (4 weeks after therapy)

## Data Availability

The dataset supporting the conclusions of this article is included within the article.

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
