# Peer review of "Efficacy and Safety of Placental Extract on Menopausal Symptoms: A Systematic Review"

_nutrients, 2025, doi:10.3390/nu17243857_

Round 1
Reviewer 1 Report
Comments and Suggestions for Authors
Esteemed authors and editorial team,
This systematic review addressing the issues of alternative therapies for alleviating the symptoms of menopause is welcome in the context of this topic which is always of interest and has been the subject of important changes of perspective.
The article is well written, I especially like the fact that the authors underline the short timeframe of treatment administration and the impossibility to generalize results, since these products have only been used in Asian countries.
I suggest the authors should comment more extensively than “Unlike conventional hormone therapy, which is associated with increased cardiovascular risks, 356 placental extracts showed no adverse cardiovascular effects” at line 355.
It is quite clear that extended use of systemic HT should not be confused with age at the initiation of systemic HT (>60 years of or >10 years after menopause onset), which was shown to be the modulating factor between a protective versus aggressive factor for CHD risk.
There are several English language faults, please carefully reread the article, I will only mention
Line 362 – aboutan
Line 368 - for having weighting
Author Response
Attached file.

Reviewer 2 Report
Comments and Suggestions for Authors
Dear authors,
The manuscript “Effects of Placental Extract on Menopausal Symptoms: a Systematic Review” addresses a highly relevant and clinically significant topic: the efficacy and safety of placental extract for menopausal symptoms.
However, several areas require clarification, reconciliation, and minor grammatical corrections to elevate the manuscript to a high standard for publication.
Please read the following file for details.
Respectfully
The Reviewer

Author Response
Attached file.

Reviewer 3 Report
Comments and Suggestions for Authors
paper title - Effects of Placental Extract on Menopausal Symptoms: a Systematic Review
here are some comments - suggestions, and questions
abstract is concise and clear, paper follow PRISMA guidelines, searched PubMed, Embase, and Scopus (2004–2024), and focused on RCTs using porcine or human placenta extracts - why 2004 to 2024 - this should be noted
a timely study with the first systematic review on placental extracts with solid method noted on PROSPERO and used PICO clearly
some issue to consider
- as i read, most studies are within an Asian context - which might limit global applicability – symptoms vary by race/ethnicity, while also the follow up seems short (up to 24 weeks)
long term safety or effects might not be explored
also some outcome/result gap might be noted - many focus on socres and physical symptoms is good, but secondary ones such as psych, aesthetics - get less detail - clarification might be needed
RoB assessment shows "some concerns" in areas like adherence (63.6% low), but supplemental figures aren't described much - would suggest to further elaborate and maybe provide or include a summary table of RoB per domain (this would clear things up)
Author Response
Attached file.
